# Lack of Interventional Studies on Suicide Prevention among Healthcare Workers: Research Gap Revealed in a Systematic Review

**DOI:** 10.3390/ijerph192013121

**Published:** 2022-10-12

**Authors:** Soo-Hyun Nam, Jeong-Hyun Nam, Chan-Young Kwon

**Affiliations:** 1Department of Nursing, Hallym Polytechnic University, Chuncheon-si 24210, Korea; 2Department of Korean Medicine, School of Korean Medicine, Pusan National University, Yangsan 50612, Korea; 3Department of Oriental Neuropsychiatry, College of Korean Medicine, Dong-Eui University, 52-57 Yangjeong-ro, Busanjin-gu, Busan 47227, Korea

**Keywords:** suicide, mental health, healthcare workers, psychosocial intervention

## Abstract

Addressing the mental health needs of healthcare workers (HCWs), who are at high risk of suicide, is an important public health issue. Therefore, this systematic review investigated the effect of psychosocial intervention targeting suicidal behavior (i.e., suicidal ideation, attempt, or fulfillment) of HCWs. Five electronic databases were searched for interventional studies reporting HCWs’ suicidal behavior outcomes. Only two interventional studies were included in this review, and no consistent conclusion was drawn from the existing literature regarding the psychosocial prevention strategies focusing on the suicide risk of HCWs. The results indicate that compared with numerous observational studies reporting poor mental health and/or severity of suicidal risk among HCWs, intervention studies using psychosocial strategies to reduce the risk of suicide are relatively scarce. Although the insufficient number and heterogeneity of the included studies leave the results inconclusive, our findings emphasize the need to fill the research gap in this field. The causes of the gap are further explored, and suggestions for future research are provided.

## 1. Introduction

The high suicide rate among healthcare workers (HCWs) is a global phenomenon. Suicide risk among HCWs may be associated with a heavy workload, stress of emergency situations, burnout, and easy access to means of suicide. Women, anesthesiologists, psychiatrists, general practitioners, and general surgeons are particularly at risk [1]. Moreover, the COVID-19 pandemic is expected to exacerbate the suicide risk of HCWs, and addressing the mental health needs of HCWs in this crisis is considered an important public health issue [2]. An observational study conducted in Australia found that during the second wave of the COVID-19 pandemic, 10.5% of HCWs reported thoughts of suicide or self-harm [3].

Given HCWs’ common mental health risks and importance to healthcare, it is critical to screen and manage the risk of suicide and to take preemptive preventive care in the workplace, i.e., healthcare settings. However, there has been no attempt to comprehensively investigate psychosocial prevention strategies focusing on the suicide risk of HCWs. Therefore, in this systematic review, psychosocial-intervention studies targeting the suicidal behavior (i.e., suicidal ideation, attempt, or fulfillment) of HCWs were collected and critically evaluated.

## 2. Materials and Methods

### 2.1. Study Search

Two researchers (SH Nam and JH Nam) searched the following five electronic bibliographic databases from June to July 2022: MEDLINE via PubMed, EMBASE via Elsevier, the Cochrane Central Register of Controlled Trials, Cumulative Index to Nursing and Allied Health Literature via EBSCO, and PsycINFO. All studies published up to the date of the search were screened. The protocol of this review was registered in the Open Science Framework registries (osf.io/2vue4). There were no changes to the methodology after the protocol registration. This systematic review complies with the PRISMA 2020 statement [4].

### 2.2. Inclusion Criteria

The inclusion criteria of this systematic review are as follows:(1)Types of study: This review included any type of original clinical interventional study, including randomized controlled clinical trials (RCTs), non-randomized controlled clinical trials (CCTs), and before–after studies. However, retrospective and observational studies were excluded. No restrictions were imposed on the language or status of publication.(2)Types of participant: Any type of HCW, including physicians and nurses, was included. No restrictions were imposed on the participants’ sex, age, or ethnicity.(3)Types of intervention: Any type of psychological intervention was included. Cognitive behavior therapy (CBT), acceptance and commitment therapy, dialectical behavior therapy, meditation, mindfulness-based intervention, autogenic training, yoga, tai chi, qigong, breathing exercises, music therapy, guided imagery, and biofeedback were regarded as psychological interventions of interest.(4)Types of control: For studies with controls, including RCTs and CCTs, there was no restriction on control intervention. That is, we allowed studies that involved no treatment, wait-list, treatment as usual, placebo or sham treatment, attention control, and active comparators.(5)Primary outcome: Any validated measure of suicidal ideation, such as the Beck scale for suicidal ideation [5], was considered the primary outcome. Secondary outcomes included any other measure of suicidal behavior, including suicidal ideation, attempt, or fulfilment.

### 2.3. Study Selection

In the first screening process, two independent researchers (SH Nam and JH Nam) screened the titles and abstracts of the searched documents to identify potentially related articles. Then, the same researchers independently reviewed the full texts of the screened studies. The final inclusion of studies was decided through the two-step screening process. In the screening process, any disagreement between the researchers was resolved through discussion. EndNote X20 (Clarivate Analytics, Philadelphia, PA, USA) was used to manage the quotations of the included articles. The study-selection process is presented as a PRISMA flow diagram.

### 2.4. Risk of Bias Assessment

Two independent reviewers (SH Nam and JH Nam) carried out a risk-of-bias (RoB) assessment performed using the Cochrane Collaboration’s RoB 2 tool [6], because the included articles were all RCTs. The tool included the following criteria: bias arising from the randomization process, bias due to deviations from the intended interventions (the effect of assignment to an intervention and the effect of starting and adhering to an intervention), bias due to missing outcome data, bias in the measurement of the outcome, and bias in the selection of the reported results [6]. Subsequently, overall bias was determined based on the results of the items. Each item was rated as “low risk,” “some concern,” or “high risk” [6]. Disagreement regarding the study selection was resolved through discussion by the two reviewers (SH Nam and JH Nam); if the discrepancies could not be resolved, the third author (CY Kwon) intervened.

### 2.5. Data Extraction

Data extraction was conducted by two independent reviewers (SH Nam and JH Nam) utilizing a pre-defined extraction form. For the included articles, the following information was extracted: first author’s name, year of publication, country of the first author, sample size, mean age, type of HCWs, pathological condition of HCWs, interventions used, control interventions, intervention period, measurements, suicide outcomes, and results of the study. Disagreement regarding data extraction was resolved through discussion by the two reviewers, and if the disagreement could not be resolved, the third author (CY Kwon) intervened.

### 2.6. Data Analysis

Qualitative analysis was used for all the included studies based on suicide outcomes. Quantitative synthesis (i.e., meta-analysis) was not performed since sufficient homogeneous data from the RCTs did not exist. Additionally, for the same reason, it was difficult to evaluate publication bias through funnel-plot generation.

## 3. Results

### 3.1. Study Search

Of the 5006 documents initially searched, the titles and abstracts of 3626 were screened after excluding duplicate articles. During the initial screening process, 3576 unrelated documents were excluded. The full texts of the remaining 50 articles were then reviewed. Of them, we excluded 48 articles since they did not meet the population criteria (n = 4), did not present sufficient outcome data (n = 11), were not interventional studies (n = 31), or only included a study protocol (n = 2). Finally, only two studies [7,8] were included in this review (Figure 1).

### 3.2. Characteristics of Included Studies

Both of the included studies [7,8] were RCTs, and one study [7] was conducted in the U.S., while the other [8] was conducted in Australia. Their sample sizes were 21 [8] and 199 [7], respectively, and their types of HCWs were medical interns [7] and junior doctors [8]. Guille et al. [7] reported the clinical conditions of HCWs, which included a history of depression and current suicidal ideation. However, Taylor et al. [8] did not report the clinical conditions of the participants. According to Guille et al. [7], the interventions consisted of four weekly web-based CBT sessions lasting 30 min, which aimed to facilitate cognitive restructuring techniques that promote the ability to identify overly negative thoughts or unrealistic beliefs and to adopt problem-solving strategies. Taylor et al. [8] performed weekly personalized yoga sessions lasting one hour (Table 1).

### 3.3. Methodological Qualities of Included Studies

Both included studies [7,8] provided information on allocation-sequence concealment, and the baseline difference was not significantly different between the two groups. Therefore, the two studies [7,8] were judged to have a low RoB in the domain of bias arising from the randomization process (Table 2). In Guille et al. [7], there was clear information on whether participants were aware of their assigned group during the intervention and whether there were deviations from the intended intervention beyond what is expected in usual practice. This study [7] was thus judged to have a low RoB in the domain of bias due to deviations from the intended interventions (the effect of assignment to an intervention). However, Taylor et al. [8] did not report any information on whether the participants were blind to their assigned group during the intervention, in which the authors directly performed the yoga intervention. Therefore, this study [8] was evaluated to have some concern in the domain of bias due to deviations from the intended interventions (the effect of assignment to an intervention) (Table 2). In Guille et al. [7], the intervention was implemented successfully by providing standard contents and methods regardless of the participant’s situation, and participants adhered to the assigned intervention regimen. Therefore, this study [7] was judged to have a low RoB in the domain of bias due to deviations from the intended interventions (the effect of starting and adhering to an intervention). However, Taylor et al. [8] reported that participants adhered to the assigned intervention regimen, although the intervention applied was personalized. This study [8] was thus judged to have some concern in the domain of bias due to deviations from the intended interventions (the effect of starting and adhering to an intervention). Taylor et al. [8] reported outcome data for all randomized participants; therefore, [8] was evaluated to have a low RoB due to missing outcome data. Even though Guille et al. [7] did not report outcome data for all randomized participants, the authors clarified that there were no significant demographic and clinical characteristics between the retained participants and those who did not complete within-internship follow-up assessments. Accordingly, this study [7] was also judged as having a low RoB. The method of measuring outcomes used by Taylor et al. [8] was appropriate, but participant-reported outcomes could have been influenced by the knowledge of the assigned intervention. As such, this study [8] was determined to have some concern in the measurement of the outcome data. Guille et al. [7] reported that both the participants and measurers were blinded in the intervention procedure, so it was considered to have a low RoB in the measurement of the outcome data. Both studies [7,8] were evaluated to have some concern for bias in their respective selections of their reported results, since Taylor et al. [8] did not report any information about whether the reported results were selected, and Guille et al. [7] did not analyze and report the data in accordance with pre-specified measurement plans. The overall methodological qualities of the two included studies were rated as “low concern” [7] and “some concern” [8], respectively (Table 2).

### 3.4. Main Results

Both studies measured suicidal ideation as their outcomes. Guille et al. [7] measured suicidal ideation using item 9 of the Patient Health Questionnaire-9 (PHQ-9) [9]. Suicidal ideation was considered existent if the participants answered positively to having “*thoughts that you would be better off dead, or hurting yourself*” for “*several days*”, “*more than half the days*”, or “*nearly every day*” over the past two weeks (i.e., a score of ≥1 on item 9 of PHQ-9). Taylor et al. [8] used a five-item Suicidal Ideation Attributes Scale (SIDAS) [10]. The scores of this tool > 20 indicated a high risk of suicidal behavior. According to Guille et al. [7], the CBT group generally showed significantly lower post-intervention suicidal ideation than the attention control (AC) group on the PHQ-9. The CBT group was 60% less likely to report suicidal ideation during the internship year compared with those assigned to the AC group (risk ratio (RR): 0.40, 95% confidential intervals 0.17–0.91; *p* = 0.03). Taylor et al. [8] calculated between-group differences and found no significant difference between the yoga group (pre 9 ± 3, post 11 ± 1) and fitness group (pre 11 ± 6, post 10 ± 4) in SIDAS scores (z = 0.72, *p* = 0.47).

## 4. Discussion

This review comprehensively and critically investigated the effects of various psychosocial interventions on suicidal behavior among HCWs. However, the findings of this review highlight the lack of research currently available on this topic. Comparing the findings of this review, numerous observational studies [1,2,3,11,12] reported poor mental health and severity of suicidal risk among HCWs. However, studies using psychosocial intervention to reduce the risk of suicide have been poorly conducted. This problem has been pointed out [13], but there seems to be no satisfactory progress so far. This research gap emphasizes the need for interventional research in this field. The findings of this review were compared with a previous review by Dutheil et al. [1] to highlight the research gap in this field (Table 3).

In order to fill this research gap, the cause of the gap should be explored. First, one of the biggest challenges associated with examining HCWs’ mental health is the stigma associated with mental illness, which creates barriers to seeking help, and is pervasive in the medical profession [14,15]. This stigma may have made it difficult for HCWs to seek help for mental health problems or participate in relevant clinical trials. To facilitate intervention research in this field, efforts are needed to reduce the stigma associated with mental health issues among HCWs. In addition, in order to minimize the impact of this stigma, group interventions conducted at hospital units or community units may be more appropriate than individualized interventions. Moreover, as in the case of Guille et al. [7], clinical research that applies web-based interventions and guarantees anonymity can be a promising clinical research strategy in this field. Second, due to the nature of the job, which requires saving lives in urgent situations and caring for emergency patients, it is difficult for HCWs to take care of their own mental health, including their emotions and needs. Especially in the COVID-19 pandemic era, in which supplies and staffing are extremely insufficient, there is little room to evaluate mental health, as HCWs deal with a demanding amount of work and care for patients [16]. Furthermore, HCWs are often forced to hide their own needs and feelings despite constantly being subjected to stressful situations [17], which prevents them from closely monitoring their mental health. As a result, they may put off seeking treatment for their emotional state, and this might lead to delayed treatment intervention. Therefore, it is important to manage the mental health of HCWs in a timely manner, such as by periodically conducting mental health screenings at the medical-institution level. Furthermore, it may be necessary to open a psychological counseling window secured with anonymity for HCWs and provide interventions targeting the maintenance of psychological well-being in a preventive way. Third, considering the reality of HCWs’ irregular work schedules and high workload, it is realistically impossible to participate in the traditional long-term psychological interventions, especially in the case of interventions conducted at medical institutions. In order to increase the accessibility of interventions, the following strategies are needed. Brief therapeutic interventions were also found to be effective [18]; therefore, it is necessary to decide upon the number and duration of sessions efficiently, considering the characteristics of the organization and the workload of the healthcare institution. Moreover, it is crucial to subdivide the intervention time by reflecting upon the individual’s schedule as much as possible to increase the accessibility of the intervention, considering the discontinuity of the HCW’s work. Finally, adopting digital technologies such as E-health, smartphone apps, and tele-health interventions—which have been proved to be effective [19,20,21,22]—might be a good alternative in situations where it is difficult to perform face-to-face interventions.

The limitations of this review are as follows. First, due to the insufficient number and heterogeneity of the included studies, we could not perform a meta-analysis on suicidal behavior among HCWs. When sufficient research is gathered, future studies may conduct a quantitative analysis to determine the effectiveness of psychological interventions for suicidal behavior of HCWs. Second, given that suicidal ideation was self-reported, which implies that the results may be underestimated, it is possible that the HCWs did not honestly respond since they may have been afraid of the stigma [14,15]. Furthermore, it is worth noting the possible involvement of the social desirability bias that HCWs should not have suicidal thoughts. Therefore, actual suicidal thoughts or behaviors may be higher than those measured in the included studies. Third, the evaluation tools to measure suicidal behaviors used in the included studies varied. The use of various measurements suggests that in future clinical studies, in order to demonstrate the effectiveness of psychological interventions for reducing suicidal behaviors, well-validated outcome measurements should be used.

## 5. Conclusions

This systematic review investigated the effect of psychosocial intervention targeting suicidal behavior (i.e., suicidal ideation, attempt, or fulfillment) of HCWs. Only two interventional studies were included in this review, and no consistent conclusion could be drawn from the studies regarding the psychosocial strategies to reduce the suicide risk of HCWs. Our results indicate that compared with the numerous studies that have reported poor mental health and severity of suicidal risk among HCWs, interventional studies using psychosocial methods to reduce the risk of suicide have been scarcely conducted. Based on the causes of this research gap which we elaborated, the findings of this study may emphasize the need to fill the research gap and develop a practical intervention suitable for HCWs’ environment.

## Figures and Tables

**Figure 1 ijerph-19-13121-f001:**
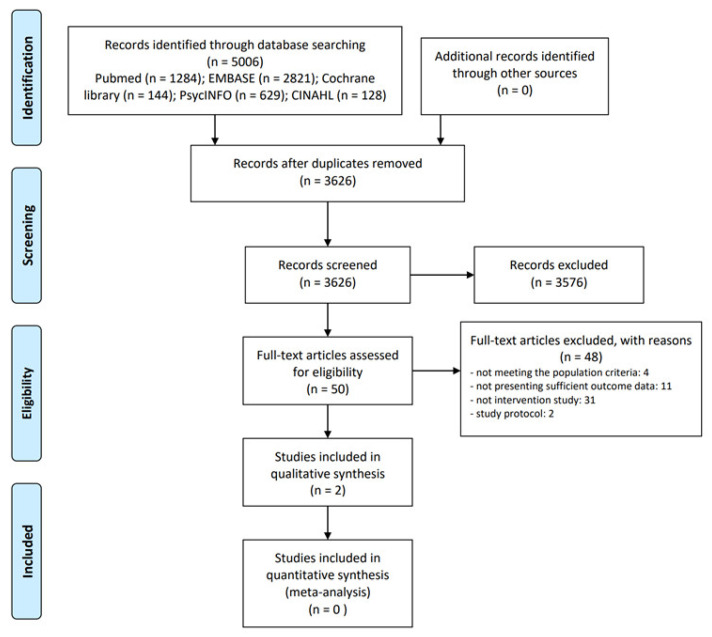
PRISMA flow diagram.

**Table 1 ijerph-19-13121-t001:** Characteristics of included studies.

Study (Reference)	Country	Sample Size	Mean Ages Mean ± SD (Range)	Type of HCWs	Pathological Condition of HCWs	Interventions Used	Control Interventions	Intervention Period
Guille [7]	U.S.	1. CBT (n = 100) 2. AC (n = 99)	1. CBT: 24.9 ± 8.7 2. AC: 25.4 ± 7.4	Medical interns	History of depression (n, %) 1. CBT: 48 (48.0) 2. ACG: 47 (47.5) Current suicidal ideation (n, %) 1. CBT: 3 (3.0) 2. ACG: 3 (3.5)	30 min web-based CBT, once a week	email, once a week	4 weeks
Taylor [8]	Australia	1. Yoga (n = 11) 2. Fitness (n = 10)	1. Yoga: 30 ± 5 (23 to 37) 2. Fitness: 30 ± 4 (26 to 37)	Junior doctors	NR	1 h personalized yoga, once a week	45–60 min group fitness, once every 2 weeks	8 weeks

Abbreviations. AC, attention control; ACG, attention-control group; CBT, cognitive behavior therapy; HCW, healthcare worker; NR, not reported, SD, standard deviation.

**Table 2 ijerph-19-13121-t002:** The risk of bias in the included studies.

Study	Q1	Q2	Q3	Q4	Q5	Q6	Q7
Guille et al. [7]	L	L	L	L	L	SC	L
Taylor et al. [8]	L	SC	SC	L	SC	SC	SC

Note. Q1: Bias arising from the randomization process; Q2: Bias due to deviations from the intended interventions (the effect of assignment to an intervention); Q3: Bias due to deviations from the intended interventions (the effect of starting and adhering to an intervention); Q4: Bias due to missing outcome data; Q5: Bias in measurement of the outcome; Q6: Bias in the selection of the reported result; and Q7: Overall bias. Abbreviations. L, low risk of bias; SC, some concern.

**Table 3 ijerph-19-13121-t003:** Research gap in the suicide risk of healthcare workers.

Research Gap	Previous Review by Dutheil et al., (2019)	Current Review by Nam et al., (2022)
Scope of review	(1) Population: healthcare workers including physicians, dental surgeons, and nurses (2) Intervention: not applicable (3) Comparator: not limited (4) Outcome: suicide attempt or suicidal ideation (5) Study type: observational studies	(1) Population: healthcare workers including physicians, medical interns, and nurses (2) Intervention: psychological interventions (3) Comparator: no treatment, wait-list, treatment as usual, placebo, attention control, and active comparators (4) Outcome: suicidal behaviors (suicidal ideation, attempt, or fulfillment) (5) Study type: intervention studies
Database searched	PubMed, Cochrane Library, Science Direct and EMBASE	PubMed, EMBASE, the Cochrane Central Register of Controlled Trials, Cumulative Index to Nursing and Allied Health Literature, and PsycINFO
Search period	Inception to April 2019	From June 2022 to July 2022
Included studies	N = 61	N = 2
Main findings	The overall SMR for suicide in physicians was 1.44 (95% CIs 1.16, 1.72), with an important heterogeneity (I^2^ = 93.9%, *p* < 0.001). Females were at higher risk (SMR = 1.9; 95% CIs 1.49, 2.58; and ES = 0.67; 95% CIs 0.19, 1.14; *p* < 0.001) compared with males. U.S. physicians were at higher risk (ES = 1.34; 95% CIs 1.28, 1.55; *p* < 0.001) than those in other parts of the world. Suicide rates decreased over time, especially in Europe (ES = −0.18; 95% CIs −0.37, −0.01; *p* = 0.044). Some specialists might be at higher risk, such as anesthesiologists, psychiatrists, general practitioners, and general surgeons. There were 1.0% (95% CIs 1.0, 2.0; *p* < 0.001) of suicide attempts and 17% (95% CIs 12, 21; *p* < 0.001) of suicidal ideation in physicians.	Only two studies were identified. According to Guille et al. [7], the CBT group generally showed significantly lower post-intervention suicidal ideation than the AC group on the PHQ-9. The CBT group was 60% less likely to report suicidal ideation during the internship year than those assigned to the AC group (RR = 0.40; 95% CIs 0.17, 0.91; *p* = 0.03). Taylor et al. [8] calculated between-group differences and found no significant difference between the yoga group (pre 9 ± 3, post 11 ± 1) and fitness group (pre 11 ± 6, post 10 ± 4) in SIDAS scores (z = 0.72, *p* = 0.47).
Conclusions	Physicians have an at-risk profession of suicide, and females are particularly at risk. The rate of suicide among physicians decreased over time, especially in Europe. The high prevalence of physicians who attempted suicide, as well as those with suicidal ideation, should benefits from preventive strategies in the workplace.	No consistent conclusion can be drawn from the existing literature regarding psychosocial prevention strategies focusing on the suicide risk of HCWs. Intervention studies using psychosocial intervention to reduce the risk of suicide among HCWs are relatively scarce.

Abbreviations. AC, attention control; CBT, cognitive behavior therapy; CIs, confidential intervals; ES, effect size; HCW, healthcare worker; PHQ-9, the Patient Health Questionnaire-9; RR, risk ratio; SIDAS, the Suicidal Ideation Attributes Scale; SMR, standardized mortality ratio.

## Data Availability

This data used to support the findings of this study are included within the article.

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
