# Peer review of "Lack of Interventional Studies on Suicide Prevention among Healthcare Workers: Research Gap Revealed in a Systematic Review"

_ijerph, 2022, doi:10.3390/ijerph192013121_

Round 1

Reviewer 1 Report

I was pleased to read the manuscript entitled "Lack of interventional studies on suicide prevention among healthcare workers: Research gap revealed in a systematic review" and to review it.

The present review explored the current status of literature regarding the impact of various psychosocial interventions on suicidal behaviour among healthcare workers (HCW). Unfortunately, the authors in their extensive search of the literature have found only two studies that examined the effects of psychosocial interventions on suicidal behavior among HCWs. This shows a deep gap in this field of research. On the other hand, it is questionable whether it was worthwhile to review only two studies. However, these authors provide a "comprehensive and critical" overview of these studies.

The review is written professionally, following the requirements that are set for reviews of this type. Therefore, I have only a few minor comments.

1. Discussion: I question whether it is worth presenting Table 3. This table compares the review conducted by the authors with a review conducted by another group of authors, which evaluates completely different types (observational) of studies.

2. Small corrections: Table 1 -explain abbreviation ACG; Table 2 - explain abbreviation Q4.

Thank you for considering my opinion. I encourage authors to keep on working to improve the manuscript.

Author Response

  • Response to Comments from Reviewer 1

Overall comment:

I was pleased to read the manuscript entitled "Lack of interventional studies on suicide prevention among healthcare workers: Research gap revealed in a systematic review" and to review it.

The present review explored the current status of literature regarding the impact of various psychosocial interventions on suicidal behaviour among healthcare workers (HCW). Unfortunately, the authors in their extensive search of the literature have found only two studies that examined the effects of psychosocial interventions on suicidal behavior among HCWs. This shows a deep gap in this field of research. On the other hand, it is questionable whether it was worthwhile to review only two studies. However, these authors provide a "comprehensive and critical" overview of these studies.

The review is written professionally, following the requirements that are set for reviews of this type. Therefore, I have only a few minor comments..

Response:

Thank you for your careful review and insightful comments that have significantly enhanced our manuscript.

Comment 1:

Discussion: I question whether it is worth presenting Table 3. This table compares the review conducted by the authors with a review conducted by another group of authors, which evaluates completely different types (observational) of studies.

Response 1:           

Thank you for your comments. Despite the fact that differences exist between the two reviews regarding study design of interest (observational vs. intervention studies) in Table 3, it could be an important background for recognizing the ‘research gap’ what we want to emphasize in this manuscript. We hope to keep Table 3 for this reason. However, if the reviewer still think that the table is unnecessary, we are willing to delete it in the next revision process.

Comment 2:

Small corrections: Table 1 -explain abbreviation ACG; Table 2 - explain abbreviation Q4.

Response 2:           

Thank you for your comments. We have added explanations, as follow.

“Abbreviations. AC, attention control; ACG, attention-control group; CBT, cognitive behavior therapy; HCW, healthcare worker; NR, not reported, SD, standard deviation.”

(Please refer red words in page 4.)

“Note. Q1: Bias arising from the randomization process; Q2: Bias due to deviations from the in-tended interventions (the effect of assignment to an intervention); Q3: Bias due to deviations from the intended interventions (the effect of starting and adhering to an intervention); Q4: Bias due to missing outcome data; Q5: Bias in measurement of the outcome; Q6: Bias in the selection of the reported result; and Q7: Overall bias”

(Please refer red words in page 5.)

Reviewer 2 Report

Review of the paper: “Lack of interventional studies on suicide prevention among healthcare workers: Research gap revealed in a systematic review”.

In my opinion:

·      the title of the article is proper  and it provides sufficient information for a potential reader;

·       the scientific problem is novel and important from the theoretical perspective and practice;

·       the Authors use the proper academic terminology; 

·        the methological rigor of the article is at sufficient level.

Author Response

  • Response to Comments from Reviewer 2

Overall comment:

In my opinion:

  • the title of the article is proper and it provides sufficient information for a potential reader;
  • the scientific problem is novel and important from the theoretical perspective and practice;
  • the Authors use the proper academic terminology;
  • the methological rigor of the article is at sufficient level.

Response:

Thank you for your careful review.

Reviewer 3 Report

The verb "commit" to described suicidal death or attempts implies criminal behavioral as in "commit a crime". It is more widely accepted to use died by or attempted suicide.

Author Response

  • Response to Comments from Reviewer 3

Overall comment:

The verb "commit" to described suicidal death or attempts implies criminal behavioral as in "commit a crime". It is more widely accepted to use died by or attempted suicide.

Response:

Thank you for your careful review. We have made revisions on the use of the verb “commit” in the Table 3.

“Physicians have an at-risk profession of suicide, and females are particularly at risk. The rate of suicide among physicians decreased over time, especially in Europe. The high prevalence of physicians who attempted to commit suicide, as well as those with suicidal ideation, should benefits from preventive strategies in the workplace.”

(Please refer red words in page 6.)
